# Exposure to a Farm Environment during Pregnancy Increases the Proportion of Arachidonic Acid in the Cord Sera of Offspring

**DOI:** 10.3390/nu11020238

**Published:** 2019-01-22

**Authors:** Malin Barman, Karin Jonsson, Agnes E. Wold, Ann-Sofie Sandberg

**Affiliations:** 1Food and Nutrition Science, Department of Biology and Biological Engineering, Chalmers University of Technology, SE-412 96 Gothenburg, Sweden; karin.jonsson@chalmers.se (K.J.); ann-sofie.sandberg@chalmers.se (A.-S.S.); 2Department of Obstetrics and Gynecology, Institute of Clinical Sciences, University of Gothenburg, SE-416 85 Gothenburg, Sweden; 3Department of Infectious Diseases, Institute of Biomedicine, University of Gothenburg, SE-405 30 Gothenburg, Sweden; agnes.wold@microbio.gu.se

**Keywords:** farming, cord blood, fatty acids, arachidonic acid, adrenic acid

## Abstract

Growing up in a farm environment is protective against allergy development. Various explanations have been put forward to explain this association. Fatty acids are regulators of immune function and the composition of fatty acids in the circulation system may affect immune development. Here, we investigate whether the fatty acid composition of cord serum differs for infants born to Farm (*n* = 26) or non-Farm mothers (*n* = 29) in the FARMFLORA birth-cohort. For comparison, the levels of fatty acids in the maternal diet, serum and breast milk around 1 month post-partum were recorded. The fatty acids in the cord sera from infants born to Farm mothers had higher proportions of arachidonic acid (20:4 *n*-6) and adrenic acid (22:4 *n*-6) than those from infants born to non-Farm mothers. No differences were found for either arachidonic acid or adrenic acid in the diet, samples of the serum, or breast milk from Farm and non-Farm mothers obtained around 1 month post-partum. The arachidonic and adrenic acid levels in the cord blood were unrelated to allergy outcome for the infants. The results suggest that a farm environment may be associated with the fatty acid composition to which the fetus is exposed during pregnancy.

## 1. Introduction

Infants who are growing up on small dairy farms are far less likely to become allergic than infants who are born in cities or in rural environments, but not on farms [1,2]. Several hypotheses have been put forward to explain the ways in which the development of the immune system of an infant can be affected by a farm environment, including exposure to microbial products and/or the different diets of farming and non-farming families. For example, cord blood cells from newborn infants born to mothers who live on dairy farms (hereinafter referred to as ‘Farm mothers’) are more likely to produce cytokines such as tumor necrosis factor (TNF)-α and interferon (IFN)-γ when stimulated ex vivo [3], suggesting that the farm environment provides immune priming signals already before birth.

The fatty acid composition of the typical farming diet differs from that of the general populations in that it contains more saturated fats and less unsaturated fats [1]. Since fatty acids exert immunomodulatory effects [4], this might also affect the early immune maturation and subsequent development of allergies. Accordingly, we have previously shown that a high proportion of poly-unsaturated fatty acids (PUFAs) among the serum phospholipids at birth correlates with a high risk of allergy development [5]. In addition, the consumption of margarine, which is lower among farming families, has been linked to a high risk of allergy development in several studies [1,6]. However, in a previous publication, we showed that the fatty acid patterns in sera sampled at 4 months of age were essentially similar between infants born into farming and non-farming families [7].

During development, the fetus is exposed to fatty acids deriving from the maternal circulation and transferred via the placenta. Saturated and mono-unsaturated fatty acids are assumed to be transferred via diffusion or a passive carrier [8], while polyunsaturated fatty acids are transported actively against a concentration gradient [9], probably to meet the large demand for these fatty acid during fetal growth. In theory, the fatty acid milieu during fetal life could be crucial to early immune development. The fatty acid milieu that affects the fetal immune system during the last part of pregnancy can be assessed by measuring the fatty acid pattern in the cord blood of the newborn infant.

In the present study, we analyzed the fatty acid compositions of the cord bloods of newborn infants from both farming and non-farming families. The aim was to investigate whether the fatty acid composition in the cord blood differed between infants born to Farm and non-Farm mothers. Secondly, the composition of the fatty acids in the cord serum were correlated to that of corresponding maternal serum obtained around 1 month post-partum, as well as maternal diet and breast milk at 1 month post-partum.

## 2. Subjects and Methods

### 2.1. Subjects

The FARMFLORA birth-cohort study comprises 65 families in the Skaraborg County in southwest Sweden. Of these, 28 run small dairy farms, while 37 live in the same rural areas, but not on farms. Pregnant women were recruited during visits to the maternity clinic between September 2005 and May 2008. Children born at gestational Week 36–42 were included. Subjects from farming milieus other than dairy farms were excluded. The study was approved by the Regional Ethics Committee in Gothenburg (No. 363–05) and written informed consent was obtained from both parents.

### 2.2. Sampling of Blood and Breast-Milk

Blood was sampled from the umbilical cord from 54 of the 65 infants directly after delivery. From the mothers, venous blood and breast milk were collected around 1 month post-partum. The blood was left at room temperature to clot for 30 min and then centrifuged. After centrifugation, the serum was removed, aliquoted, frozen, and stored at −80 °C until analyzed. Maternal blood was obtained from 54 of the mothers, and breast milk was collected from 53 of the mothers.

For sampling breast milk, the women were instructed to express 5–10 mL of breast milk manually or with a breast milk pump. The breast milk was collected at home by the mother during the second breast-feeding of the day. The mothers were instructed to freeze immediately the samples at −20 °C at home in sterile plastic tubes. Study nurses collected the samples within 6 months of sampling and transferred the tubes to storage at −80 °C, where they were stored until analyzed.

### 2.3. Analysis of Phospholipid Fatty Acids in Serum

The maternal and infant fatty acid compositions were analyzed for the phospholipid fraction as a percentage of the total fatty acids. As the phospholipid fraction is considered to have a slower rate of turnover than the free fatty acids in serum, it reflects the fatty acid milieu over a period of several weeks. Initially, the serum (500 µL) was mixed with 4 mL of a chloroform:methanol (1:2) mix for extraction of the fat [10]. Then, 2 mL of 0.5% NaCl solution and 50 µL of the internal standard, heptadecanoic acid 17:0, were added. After vortexing and centrifugation, the chloroform phase was collected, and after evaporation, the fat was dissolved in 200 µL chloroform. Aminopropyl solid-phase extraction columns (Isolute NH2, 6-mL, 500-mg; IST, Hengoed, Mid Glamorgon, UK) were used for phospholipid separation [11]. The phospholipid fraction was collected and evaporated. Toluene (1 mL) and 1 mL of methanol acetyl chloride [acetyl chloride (10%) in methanol] were added to the evaporated phospholipids, to promote methylation of the fatty acids. The phospholipids were converted to methyl esters at 70 °C for 2 h. Thereafter, the fatty acid methyl esters (FAMEs) were extracted with petroleum ether and, following evaporation, dissolved in isooctane. FAMEs were analyzed in a Hewlett-Packard 5890 capillary gas chromatograph (Waldbronn, Germany) equipped with a Hewlett-Packard auto-injector 7673 (Walbronn, Germany) and detected with flame ionization detection (FID). H_2_ was used as the carrier gas. Two columns were used. For separation of 20–22-carbon-long fatty acids, a fused silica SPB-5 (30 m × 0.25 mm × 0.25 µm DF) column (Supelco, Bellefonte, PA, USA) was used; the initial temperature was 150 °C, which was increased at a rate of 4 °C/min until 320 °C, and then maintained at this temperature for 3 min. For separation of 16–18-carbon-long fatty acids, a DB-WAX (30 m × 0.25 mm × 0.25 µm d_F_) column (J&W Scientific, Folsom, CA, USA) was used; the initial temperature was 100 °C, which was increased at a rate of 4 °C/min until 250 °C, and then maintained at this temperature for 5 min. A sample of 2 µL was injected and the split was 3:1. The Borwin HPLC software (Le Fontanil, France) was used for evaluation of the HPLC profiles.

### 2.4. Analysis of Fatty Acids in Breast Milk

The fatty acids in the total lipid fraction of breast milk were analyzed by gas chromatography after conversion to methyl esters [12]. Thawed milk samples were mixed (1:1) with toluene and acetyl chloride (10%) in methanol, including 25 µg of an internal standard (heneicosanoic acid, 21:0) and incubated at 70 °C for 2 h. The methyl esters were then analyzed as described above for the serum samples.

### 2.5. Assessment of Maternal Diet during Pregnancy and Lactation

Maternal dietary intake during pregnancy was assessed using a semi-quantitative food-frequency questionnaire and has previously been reported [1]. Around 1 and 4 months post-partum (during lactation), the maternal dietary intake was assessed using a 24 h dietary recall followed by a 24 h food diary as described previously [1], and the nutrient and energy compositions of the diet were calculated based on the food composition database of the Swedish National Food Agency (Diet 32; Aivo AB, Stockholm, Sweden). The results from the dietary assessment carried out at 4 months post-partum has been reported [1].

### 2.6. Diagnosis of Allergy at 1.5 and 3.0 Years of Age

The infants in the FARMFLORA birth-cohort have been diagnosed at 18 and 36 months of age for food allergy, eczema, asthma, and allergic rhinoconjunctivitis according to carefully standardized protocols, as previously described in detail [1]. Clinical diagnoses were made by trained pediatric allergologists, and blood tests (Phadia, Uppsala, Sweden) were used to assess sensitization to inhalant allergens (Phadiatop) and common foods (6-mix food) [2]. In the statistical analyses, any type of allergy at 36 months was used as the outcome, due to the limited number of allergic children. The children who were diagnosed as being allergic at 18 months but not at 36 months of age (transient allergy) were not included in either the healthy or the allergic group

### 2.7. Statistical Analysis

Due to small sample sizes and partly non-normal and skewed distributions of the variables, nonparametric tests were used, and the data are presented as medians and interquartile ranges. Differences in serum fatty acids were tested for significance using the Mann-Whitney *U*-test. Participant characteristics were analyzed using χ^2^ test or Fisher’s exact test (when 20% or more of the cells had an expected count less than 5) for categorical variables. Linear regression models were used to adjust the association between farm environment (Farm group or non-Farm group) and the proportion of arachidonic acid in the cord serum, adjusting for potential confounders. Confounders were chosen based on differences between Farm and non-Farm families. Thus, the level of paternal education and the sex of the child were included as confounders, both separately and together. The linearity of the linear regressions was tested visually with a scatter plot of predicted values against residuals showing that the relationship was roughly linear around zero and the variance of the residuals was homogeneous across levels of the predicted values for all regression models. Also, a Q-Q plot of the residuals showed that these were normally distributed for all models. Two-tailed *p*-values of ≤0.05 were considered statistically significant. The analyses were performed with the IBM SPSS Statistics ver. 19 software (IBM Corporation, New York, NY, USA).

## 3. Results

### 3.1. Characteristics of the Study Groups

Farmers’ children were more often girls (62%), while the children of non-farmers were more often males (64%, *p* = 0.04). The newborns did not differ with respect to birth weight (3500 vs. 3600 g, *p* = 0.78), gestational age at delivery (median 40 weeks for farmers and 39 weeks for non-farmers, *p* = 0.13), or being the first-born (36% vs. 54%, *p* = 0.15). Overall, 12% (3/28) of the farmers’ infants were delivered by Cesarean section, as compared to 19% (7/37) of the controls (*p* = 0.5). There were no differences in maternal age at delivery (33 vs. 32 years, *p* = 0.46), maternal educational level (*p* = 0.2), or maternal smoking during pregnancy (0% vs. 3%, *p* = 1.0) between the Farm and non-Farm mothers. The fathers in the farming families had a lower level of education than the fathers in the non-farming families (*p* = 0.04). Education level for both the mothers and the fathers was divided into five categories: 1 = elementary school. 2 = upper secondary school 2–3 years or equivalent. 3 = qualified graduate from upper secondary engineering course. 4 = university ≤1 year. 5 = university >1 year.

### 3.2. Fatty Acid Compositions of Cord Sera from Infants Born to Farm and Non-Farm Mothers

The cord serum fatty acids were analyzed for the phospholipid fraction, with the results expressed as proportions of the total fatty acid content. Children born to Farm mothers had significantly higher proportions of arachidonic acid than infants born to rural non-Farm mothers (14% vs. 11%, *p* = 0.001) (Table 1, Figure 1A). The same was true for adrenic acid, 22:4 *n-6*, (0.74% vs. 0.66%, *p* = 0.04), total *n*-6 long-chain (LC-)PUFAs (21% vs. 18%, *p* = 0.02), and total *n*-6 PUFAs (27% vs. 25%, *p* < 0.0001, Table 1). When the values for arachidonic acid were subtracted from those for the *n*-6 PUFAs, the difference in total *n*-6 PUFA between the groups disappeared. The farmers’ children also had higher proportions of total LC-PUFAs (28% vs. 25%, *p* = 0.011) and total PUFAs (35% vs. 31%, *p* = 0.004) in their cord blood; when arachidonic acid was subtracted from the total LC-PUFAs and total PUFAs, the differences between the groups disappeared. Therefore, the differences in total LC-PUFAs and total PUFAs between the Farm and non-Farm controls were driven by the difference in the proportions of arachidonic acid.

Regarding the *n*-3 PUFAs, there was a weak and non-significant tendency towards somewhat higher proportions in the Farm infants. Cord blood levels of total *n*-6 PUFAs and total *n*-3 PUFAs correlated significantly (*rho* = 0.48, *p* < 0.001), as did cord blood levels of arachidonic acid and docosahexaenoic acid (DHA) (*rho* = 0.44, *p* = 0.001).

Mono-unsaturated fatty acids were found at slightly and insignificantly higher proportions in the non-Farm infants, while saturated fatty acids were found in equal proportions in Farm and non-Farm infants (Table 1).

### 3.3. Association between Arachidonic Acid Level in Cord Serum and Being Born into a Farm Environment, with Adjustment for Potential Confounders

Linear regression models were used to adjust the association between the level of arachidonic acid in cord serum and being born into a farm environment. The unadjusted model, with arachidonic acid as dependent variable and farming group as independent variable, showed a significant association: B = 1.99 (0.92–3.05), *p* < 0.001. Covariates that differed between the two groups were added as confounders in the adjusted model, i.e., sex of the infant and paternal education. Adjusting for sex and paternal education alone (B = 1.94 (0.832–3.041), *p* = 0.001 and B = 2.25 (1.12–3.4), *p* < 0.001) or together (B = 2.2 (1.03–3.4), *p* < 0.001) did not change the association between arachidonic acid and farm environment.

### 3.4. Associations between Arachidonic Acid and Adrenic Acid Levels in Cord Sera and Allergy Development in the Children

As previously reported, 1/28 children from a farm background (3.6%) and 10/37 control children (27%) had developed allergy at 3 years of age [1]. We examined the associations between the levels of arachidonic acid and adrenic acid in the cord sera and allergy development at 36 months of age using the Mann-Whitney *U*-test. No associations were found for arachidonic acid (median (IQR) proportion in allergic subjects: 12.7 (11.4–15.0) and in healthy subjects: 13.3 (11.3–14.6), *p* = 0.72)) or for adrenic acid (median (IQR) proportion in allergic subjects: 0.70 (0.62–0.85) and in healthy subjects: 0.71 (0.60–0.84), *p* = 0.68) and allergic diseases at 36 months of age.

### 3.5. Fatty Acid Compositions of the Maternal Serum, Breast Milk, and Diet

The proportions of fatty acids in the serum phospholipids of the maternal blood obtained around 1 month after delivery of the baby were evaluated. In general, there were no significant differences in the serum fatty acid compositions between Farm and non-Farm women (Appendix A). For example, the Farm mothers did not show higher proportions of arachidonic acid (median: 6.9%, IQR: 5.8–7.9%) than the non-Farm mothers (median: 7.2%, IQR: 6.3–8.4%, *p* = 0.14) (Figure 1A) and did not have higher proportions of adrenic acid (median: 0.29%, IQR: 0.29–0.38%) than the non-Farm control mothers (median: 0.33%, IQR: 0.27–0.38%, *p* = 0.12).

Maternal intake of fatty acids was measured 1 month post-partum using a 24 h dietary recall followed by a 24 h food diary. The Farm and non-Farm mothers had similar intake levels of arachidonic acid (median: 0.11, IQR: 0.06–0.17 g/day vs. median: 0.12, IQR: 0.08–0.14 g/day) (Figure 1B) and of the precursor fatty acid linoleic acid (median: 9.4, IQR: 6.6–12 g/day vs. median: 9.1, IQR: 6.7–12 g/day), as well as for all of the other analyzed fatty acids (Appendix A).

It is generally assumed that the PUFAs in breast milk derive from the maternal serum by passive transport/diffusion. We measured the proportions of fatty acids in the breast milk samples collected 1 month after delivery from the Farm and non-Farm women, and found them to be similar (Appendix A). Thus, the breast milk from Farm mothers did not have higher proportions of arachidonic acid (median: 0.34%, IQR: 0.32–0.43%) than the breast milk from non-Farm mothers (median: 0.39%, IQR: 0.32–0.44%, *p* = 0.64) (Figure 1C) and did not show higher proportions of adrenic acid (median: 0.07%, IQR: 0.05–0.08%) than the breast milk from non-Farm control mothers (median: 0.06%, IQR: 0.06–0.08%, *p* = 0.38).

### 3.6. Correlation between Maternal Food Intake during Pregnancy and Arachidonic Acid Levels in Cord Serum

The maternal food intake during pregnancy was assessed using a food-frequency questionnaire and has previously been reported [1]. Here, we correlated (using Spearman’s rank correlation) the proportion of arachidonic acid in the infant’s serum at birth to the mother’s dietary intake during pregnancy, focusing on foodstuffs known to be rich in arachidonic acid or the precursor fatty acid linoleic acid. The proportion of arachidonic acid in the infant’s serum at birth correlated weakly with maternal intake of full-fat cream during pregnancy (*rho* = 0.29, *p* = 0.048), and tended to correlate inversely to maternal intake of margarine (*rho* = −0.27, *p* = 0.07). However, it did not correlate to the mother’s intake of butter (*rho* = 0.17, *p* = 0.24), whole-fat milk (*rho* = 0.10, *p* = 0.52), fatty fish (*rho* = −0.21, *p* = 0.16) or pork (*rho* = −0.01, *p* = 0.95).

## 4. Discussion

We show that there are higher proportions of arachidonic acid and adrenic acid in the cord blood phospholipid fraction of infants born to Farm mothers than infants born to non-Farm mothers. We have previously shown that Farm mothers consume more butter during pregnancy than do control mothers [1]. As butter contains arachidonic acid (0.13 g/100 g), one might argue that the findings of more arachidonic acid in the infant cord blood serum at birth reflects a higher butter intake by the mother. However, we found no correlations between the level of butter intake by the mothers and the arachidonic acid proportions in the cord serum. In accordance with our findings, Donahue et al. reported that arachidonic acid in cord blood did not correlate to maternal dietary intake of arachidonic acid [13]. Although we did not have maternal sera collected during pregnancy or delivery, fatty acids could be analyzed in the maternal serum and maternal breast milk collected around 1 month after delivery of the baby. Dietary intake of fatty acids has been shown to be reflected in the measured levels in the blood and breast milk of pregnant and lactating mothers [14,15,16]. We also assessed the maternal diet 1 month after delivery, including analysis of the fatty acid composition. The Farm women neither consumed more dietary arachidonic acid or adrenic acid nor had higher serum or breast milk concentrations of these fatty acids post-partum, as compared to the non-Farm women. A limitation of the present study is that we lacked serum samples from the pregnant women. It has previously been shown that the fatty acid profiles differ between the pregnant and post-partum phases [17,18]. Thus, the associations between the infant and maternal serum fatty acid profiles should be interpreted with caution, and the finding should be considered preliminary.

Although it does not seem likely that mothers change their diet drastically between pregnancy and the post-partum, a rapid decrease in the proportions of maternal fatty acids in the phospholipid fraction post-partum have been reported [17,18]. Already at 1 month post-partum, the concentrations of arachidonic acid in the maternal sera had decreased dramatically [19]. However, we consider it unlikely that that affected in different ways the Farm and non-Farm women.

Another limitation of this study is the lack of information regarding maternal body mass index (BMI) to be added as a covariate to the regression analyses. Furthermore, the sampling of breast milk was not standardized, and since the fat content of breast milk varies with the time of day, full vs. partial expression, fore vs. hind milk etc., this could have caused some variability in the fat compositions of the breast milk samples.

Interestingly, the Farm and non-Farm infants did not differ regarding the arachidonic acid levels in their sera at 4 months of age [7]. Thus, it appears that the comparatively high levels of arachidonic acid seen in Farm infants are confined to the fetal period, as reflected in the cord blood samples. It is possible that exposure in utero to a farming milieu enhances delivery of arachidonic acid to the fetus, or reduces its consumption. Although the fetus may produce some arachidonic acid, endogenous production of LC-PUFAs during fetal life is believed to be low [20], and the fetal demand for LC-PUFAs is considered to be met chiefly by active transport across the placenta. One may speculate that the farm environment affects placental function, leading to the upregulation of LC-PUFA transport. Along these lines, we have shown that the Farm infants in the presently investigated cohort have higher levels of the B-cell-stimulating cytokine BAFF (B-cell-activating factor) in the cord blood, as compared to non-Farm infants [21]. BAFF is produced by the stromal cells in the placenta in response to various cytokines, such as IFN-γ [22], which in turn are produced during immune responses. Clearly, the fetus is exposed to environmental cues in utero.

We are not aware of any previous study that has demonstrated differences in cord blood fatty acid patterns between infants from farming and non-farming backgrounds. However, it is clear that the fetus receives environmental cues in utero and that exposure to a farm environment modulates the function of the fetal immune system [3]. Both *n*-3 and *n*-6 PUFAs are potent inhibitors of the activation of T cells [23,24,25], particularly those of the Th1 subset [26], and they inhibit IFN-γ secretion by mitogen-stimulated T cells [26,27]. One possibility is that enhanced delivery of Th1-dampening LC-PUFAs (arachidonic acid) to the fetus occurs in response to immune-activating signals and serves to modulate the immune response to avoid overt Th1-driven inflammation, which could terminate the pregnancy [28].

In the present cohort, we found no association between arachidonic acid or adrenic acid in the cord serum and allergy development. Arachidonic acid is the main long-chain *n*-6 PUFA component in phospholipid membranes and the precursor of eicosanoids, such as prostaglandins, leukotrienes, and thromboxanes [29]. Prostaglandins stimulate the maturation of dendritic cells into a Th2-promoting phenotype [30]. Prostaglandin E2 also suppresses inflammation, including allergic inflammation, via EP4 receptor activation [31] and also impairs the activation of innate lymphoid cells (ILC2s) that are involved in the initial phase of type 2 inflammation [32]. Therefore, a possible explanation for why the higher concentration of arachidonic acid seen in cord serum does not correlate with allergy development is the dual roles of prostaglandins in the regulation of allergic disorders. The small size of the cohort precludes firm conclusions regarding potential associations between cord blood fatty acid pattern and subsequent allergy development.

## 5. Conclusions

The higher proportions of arachidonic acid and adrenic acid in the cord blood phospholipid fraction of infants born to Farm mothers than infants born to non-Farm mothers suggest that an association exists between the farm environment and placental and/or fetal function, and that this association involves regulation by a potent class of immunomodulators—the LC-PUFAs. Whether this is due to increased active transport of fatty acids across the placenta, different fetal elongation capacity, reduced fetal consumption, or the endocrine function of the placenta being changed under specific metabolic conditions, remains to be elucidated.

## Figures and Tables

**Figure 1 nutrients-11-00238-f001:**
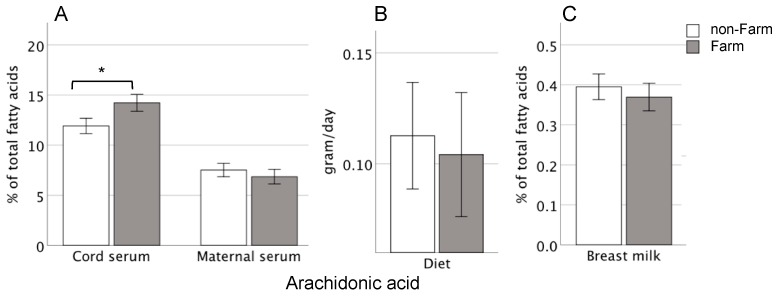
Proportions of arachidonic acid among the total fatty acids in: (**A**) cord serum obtained at delivery from infants and in maternal serum obtained post-partum, from non-farming control (open bars) or farming (filled gray bars) families. (**B**) The maternal diet 1 month post-partum; and (**C**) breast milk obtained around 1 month post-partum. Differences in the proportions of arachidonic acid between samples from Farm and non-Farm subjects were calculated using the Mann-Whitney *U*-test. Error bars are mean +/−2 SE; *: *p* < 0.05.

**Table 1 nutrients-11-00238-t001:** Proportions of fatty acids (% of total phospholipid fatty acids) in the phospholipid fraction of cord sera of infants from farming and rural non-farming families.

Fatty Acid Proportions in Infant Cord Serum; Median (IQR)			
Fatty Acid (% of Total Fatty Acids)	Farm Infants (*n* = 26)	Non-Farm Infants (*n* = 29)	*p*-Value ^a^
18:2 *n*-6 (LA)	6.3 (5.6–6.7)	6.1 (5.4–6.7)	0.76
20:3 *n*-6	5.2 (4.8–5.5)	5.1 (4.3–5.6)	0.30
20:4 *n*-6 (AA)	14 (13–15)	11 (11–14)	0.001
22:4 *n*-6	0.74 (0.66–0.88)	0.66 (0.60–0.77)	0.039
22:5 *n*-6	0.65 (0.46–0.73)	0.55 (0.41–0.78)	0.47
*n*-6 PUFA, sum	27 (25–28)	25 (23–26)	0.001
*n*-6 LCPUFA, sum	21 (19–22)	18 (17–21)	0.002
18:3 *n*-3 (ALA)	nd	nd	-
20:5 *n*-3 (EPA)	0.96 (0.85–1.2)	0.94 (0.70–1.3)	0.81
22:5 *n*-3 (DPA)	0.55 (0.41–0.66)	0.44 (0.37–0.61)	0.32
22:6 *n*-3 (DHA)	6.0 (4.5–6.6)	5.1 (4.2–6.2)	0.25
*n*-3 LCPUFA, sum	7.6 (5.8–8.6)	6.7 (5.7–8.1)	0.39
PUFA, sum	35 (33–36)	31 (29–35)	0.010
LCPUFA, sum	28 (26–30)	25 (23–28)	0.011
Ratio, AA/DHA	2.5 (2.0–3.3)	2.4 (1.9–2.8)	0.27
Ratio, *n*-6/*n*-3 PUFA	3.7 (3.1–4.7)	3.6 (3.1–4.4)	0.58
Ratio, *n*-6/*n*-3 LC-PUFA	2.9 (2.4–3.5)	2.8 (2.3–3.2)	0.28
18:1 *n*-7	2.8 (2.5–3.0)	2.9 (2.6–3.3)	0.066
18:1 *n*-9	13 (11–16)	14 (13–16)	0.13
Monounsaturated fatty acids, sum	16 (14–19)	18 (15–19)	0.098
18:0	15 (14–16)	16 (14–17)	0.36
20:0	0.62 (0.55–0.69)	0.61 (0.54–0.67)	0.58
22:0	0.95 (0.88–1.1)	1.0 (0.87–1.1)	0.71
Saturated fatty acids, sum	16 (15–18)	18 (15–18)	0.32

Proportions of fatty acids measured as % of total phospholipid fatty acids. Data are presented as medians (interquartile range) and expressed as % of total fatty acids. ^a^ Mann-Whitney *U*-test. Abbreviations: LA, linoleic acid; AA, arachidonic acid; EPA, eicosapentaenoic acid; DPA, docosapentaenoic acid; DHA, docosahexaenoic acid; PUFA, poly-unsaturated fatty acid; LC-PUFA, long-chain PUFA (i.e., 20–22-carbon-long fatty acids); nd, not detectable.

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
