# Peer review of "Exposure to a Farm Environment during Pregnancy Increases the Proportion of Arachidonic Acid in the Cord Sera of Offspring"

_nutrients, 2019, doi:10.3390/nu11020238_

Round 1

Reviewer 1 Report

Thank you for the opportunity to review this paper. The manuscript provides interesting information from an environmental perspective on how it could affect AA concentrations in infants. See comments below:

Line 46: Add acronym for PUFA here since it’s the first time you are mentioning it.

Paragraph line 49-53: Review this, clarify if this is part of the present manuscript or from a previous publication as it sounds like some results.

Line 57: You can just leave PUFA if spelling it out in line 46.

Line 47 (sampling of breast milk): Since milk collections were not standardized, this should be mentioned in the limitations. The fat content in breast milk is the most variable of all nutrients, so I’m concerned about the variability you may have gotten since you have a lot of factors that could have an impact on the fat (manual collection vs pump, full vs partial expression, fore vs hind milk, time of the day, etc.).

Line 89: It looks like you are missing a citation number.

Paragraph from line 85 to 107: Review for grammar.

Line 111: Spell out the name of the 21:0 FA.

Paragraph from line 113 to 120: I believe “assessed” is a more accurate term when referring to dietary data instead of “measured”.

Line 130: I believe table 1 is missing…?

Table 2: use capitals for both NDs. Very good to include the ratios in the table.

Line 223 – Discussion: Where there any other foods in addition to butter that could be adding to the concentrations found in maternal serum?

Lines 238 and 239: Review the format of references.

Line 252: Write down “gamma’ in the same way that was done at the beginning.

Author Response

Comments from reviewer 1: 

Thank you for the opportunity to review this paper. The manuscript provides interesting information from an environmental perspective on how it could affect AA concentrations in infants. See comments below:

We thank the reviewer for very constructive comments. Please find our comments below in red. 

Line 46: Add acronym for PUFA here since it’s the first time you are mentioning it.

Changed

Paragraph line 49-53: Review this, clarify if this is part of the present manuscript or from a previous publication as it sounds like some results. 

This sentence has been adjusted. 

Line 57: You can just leave PUFA if spelling it out in line 46.

Changed

Line 47 (sampling of breast milk): Since milk collections were not standardized, this should be mentioned in the limitations. The fat content in breast milk is the most variable of all nutrients, so I’m concerned about the variability you may have gotten since you have a lot of factors that could have an impact on the fat (manual collection vs pump, full vs partial expression, fore vs hind milk, time of the day, etc.).

This is now added as a limitation in the discussion. 

Line 89: It looks like you are missing a citation number.

Changed

Paragraph from line 85 to 107: Review for grammar. 

We have sent the whole manuscript for English editing and these sentences has been edited.  

Line 111: Spell out the name of the 21:0 FA.

Changed

Paragraph from line 113 to 120: I believe “assessed” is a more accurate term when referring to dietary data instead of “measured”.

Changed

Line 130: I believe table 1 is missing…?

Yes, we had a table 1 that we deleted. We forgot to change the number on table 2. This is now changed. 

Table 2: use capitals for both NDs. Very good to include the ratios in the table.

Changed to nd for both. 

Line 223 – Discussion: Where there any other foods in addition to butter that could be adding to the concentrations found in maternal serum?

Farming mothers had a higher intake of butter and whole fat milk. None of these correlated to the proportion of arachidonic acid in cord serum. 

Lines 238 and 239: Review the format of references.

Changed

Line 252: Write down “gamma’ in the same way that was done at the beginning.

Changed

Reviewer 2 Report

The data and methods presented in the manuscript are original and interesting regarding the increments of the proportion of arachidonic acid in cord serum of the farmers’ infants.

General Comments

The strength of this study is the use of various types of clinical samples including cord serum, maternal serum, breast milk, and diet to specify the differences of fatty acid compositions. The manuscript is, for the most part, clearly presented, and the results provide a new suggestion to understand the importance of fatty acid composition of cord serum in the regulation of allergy development. There are, however, several points that the authors need to address.

Specific Comments

 Major comments

1.       Line 262-264: The authors showed no correlations between the percentages of arachidonic acid in cord serum of farmer group and allergy development. The author’s explanation is too simple and the actual result data should be presented even in the simple text style.

2.       Line 261-262 and 264-266: The authors emphasized the importance of prostaglandins, and its metabolite, PGE2, for allergy development the text. However, PGE2 also exert anti-allergic effects in the regulation of allergic inflammation in murine models of asthma and human (Birrell MA et al, Thorax. 2015 Aug;70(8):740-7. Maric J et al, J Allergy Clin Immunol. 2018 May;141(5):1761-1773.e6., respectively). Dual roles of prostaglandins in the regulation of allergic disorders might explain the reason why higher concentration of arachidonic acid in cord serum did not correlate with allergy development, which you can raise as second possibility of this point.

 Minor comments

1.       Line 30: Please correct the word “breastmilk” to “breast milk”.

2.       Line 89: Please fill in the blank of ().

3.       Line 216: Please correct the word “ofbutter” to “of butter”.

4.       Line 238-239: Please correct 3 references to the listed numbers.

Author Response

Comments from reviewer 2: 

We thank the reviewer for very constructive comments. Please find our comments below in red. 

The data and methods presented in the manuscript are original and interesting regarding the increments of the proportion of arachidonic acid in cord serum of the farmers’ infants.

General Comments

The strength of this study is the use of various types of clinical samples including cord serum, maternal serum, breast milk, and diet to specify the differences of fatty acid compositions. The manuscript is, for the most part, clearly presented, and the results provide a new suggestion to understand the importance of fatty acid composition of cord serum in the regulation of allergy development. There are, however, several points that the authors need to address.

Specific Comments

 Major comments

1.           Line 262-264: The authors showed no correlations between the percentages of arachidonic acid in cord serum of farmer group and allergy development. The author’s explanation is too simple and the actual result data should be presented even in the simple text style.

We have now added these results in the result section. Paragraph 3.4. 

2.       Line 261-262 and 264-266: The authors emphasized the importance of prostaglandins, and its metabolite, PGE2, for allergy development the text. However, PGE2 also exert anti-allergic effects in the regulation of allergic inflammation in murine models of asthma and human (Birrell MA et al, Thorax. 2015 Aug;70(8):740-7. Maric J et al, J Allergy Clin Immunol. 2018 May;141(5):1761-1773.e6., respectively). Dual roles of prostaglandins in the regulation of allergic disorders might explain the reason why higher concentration of arachidonic acid in cord serum did not correlate with allergy development, which you can raise as second possibility of this point.

We thank the reviewer for bringing up this second possibility. We have added this to the manuscript. 

 Minor comments

1.       Line 30: Please correct the word “breastmilk” to “breast milk”.

2.       Line 89: Please fill in the blank of ().

3.       Line 216: Please correct the word “ofbutter” to “of butter”.

4.       Line 238-239: Please correct 3 references to the listed numbers.

All minor comments has been addressed. 

Reviewer 3 Report

The present manuscript describes the differences in the cord blood fatty acid composition between infants born to farming and non-farming mothers. Further, the fatty acid composition in maternal sera and breastmilk 1 month postpartum are tested for differences in farming and non-farming mothers. The authors report higher levels of n-6 PUFAs arachidonic acid and adrenic acid in cord blood of farming mothers. No differences were observed between farming and non-farming mothers’ sera, breastmilk or diet. The conclusions drawn are too strong given the observational study design and small sample size. 

1.     The authors assume causality of the observed associations and refer to influence and impact of the farming environment (starting with the title and throughout the manuscript). However, this is a small, observational and explorative study and causality cannot be implied. The wording should be changed throughout the manuscript, replacing with “association”, and the conclusion downplayed.

2.     Sample size: what was the rational for the inclusion of only 65 families when initially planning the study design? The authors should include a statement on the power calculation and its assumptions when designing the study. 

3.    Line 22: “ The fatty acid pattern in cord serum is also associated with allergy development.” The statement is too strong as there is a lot of controversy. This should be changed to reflect the level of evidence. This applies also to the introduction.

4.    The numbers in the abstract and methods section are not identical. It should be clear how many subjects are included in the study and how many were excluded for which reasons

5.    The results are often overstated. It should only be reported what was exactly tested. For example in the abstract (line 29) it says “This difference could neither be explained by higher levels of n-6 PUFAs in the diet of farming mothers nor to higher n-6 PUFA proportions in their serum or breastmilk.”. As only univariate tests of fatty acid composition in cord blood, breast milk, maternal serum and diet are conducted, it cannot be implied that effects are independent. This should be reworded saying something like “differences in AA found in cord blood, but not in maternal blood, breast milk or diet.”

6.    Allergies: The authors state in the abstract and discussion that allergy outcomes were not associated with fatty acid composition in cord blood. It is not mentioned and defined in the methods and results section, nor are any results presented. It is not clear which allergies, assessed how and at what time points, prevalences etc. The authors should properly analyse and present the association with allergies or delete the relevant sections in abstract and discussion.  

7.    Line 75: how was tested if the cord blood sample was contaminated by maternal blood?

8.    Line 113: it seems different methods have been used for the assessment of maternal diet during and after pregnancy. It is not clear, why the methods were changed as this does not allow any comparisons between time points. However, as only the diet during pregnancy was further analysed, the methods section should only focus on this assessment.

9.    Line 124: “Wilcoxon signed rank test was used to compare maternal and infant fatty acid proportions” It is not clear where these results are presented. It would also be more clear if it would be specified which maternal measurements are meant and “cord blood fatty acid proportion” used consistently rather than “infant fatty acid proportion”

10.  Line 126: it is not clear when the chi2 test was applied, as all the variables seem binary or continuous.

11.  Line 127: in line 122, it is explained that the variables are skewed and non-normally distributed, and therefore, non-parametric tests are applied (which is correct). Later, linear regression analysis is conducted, which is a parametric test and very prone to outliers. It should be included how the assumptions of the linear regression model were tested and if they hold. 

12.  Line 130: Table 1 is missing

13.  Line 140: it is reported that “Eleven percent (11/28)” farm children were deliverd by c-section. The numbers should be double checked as 11/28 would be ~40%.

14.  Table 2: the effects of PUFA sum, LCPUFA sum as well as n-6 PUFA sum and n-6 LC-PUFA sum are probably driven by AA. This should be stated at some point.

15.  Line 178: “We have previously assessed the fatty food intake during pregnancy in the cohort [1]. Since farming mothers ate more butter, which contains some arachidonic acid, we performed linear regression model with maternal butter intake as confounder.” This statement should be included in the methods section but not results. It is also not clear if the model includes only maternal butter intake during pregnancy or additionally sex and paternal education

16.  Line 194: the supplementary tables should be presented in order of appearance in the main text

17.  Figure 1: change  in“displayed as +/- 2 standard errors” to“mean +/- 2SD” in the legend

18.  Line 211: “infant’s proportion… ” should be described more precisely as “proportion .. in cord blood”

19.  Was also assessed whether the farm mother consumed (unprocessed) farm milk during pregnancy?

20.  Line 226: “In accordance with our findings, Donahue et al. reported strong correlations for EPA and DHA for both diet and cord plasma and maternal blood and cord plasma.”  The correlations of EPA and DHA are not reported in the present manuscript therefore, the statement “in accordance with our findings” should be deleted

21.  Line 234: “nor did they have higher serum or breast milk concentrations of these fatty acids in serum or breast-milk one month post-partum” should be corrected

22.  Lines 238, 239: references are not formatted correctly and should be corrected

23.  When discussing the potential mechanisms in the discussion section, it would be helpful to the reader to summarize all theories in one paragraph and distinguish clearly between n-6 and n-3 PUFA rather than reporting PUFA in general. This is confusing as n-6 PUFA are assumed to have a pro-inflammatory effect, as stated in line 260. The speculation on underlying mechanisms in the conclusion should be deleted (lines 274-276)

Author Response

Answers to the reviewers comments: Report 3

The present manuscript describes the differences in the cord blood fatty acid composition between infants born to farming and non-farming mothers. Further, the fatty acid composition in maternal sera and breastmilk 1 month postpartum are tested for differences in farming and non-farming mothers. The authors report higher levels of n-6 PUFAs arachidonic acid and adrenic acid in cord blood of farming mothers. No differences were observed between farming and non-farming mothers’ sera, breastmilk or diet. The conclusions drawn are too strong given the observational study design and small sample size.

We thank the reviewer for going through the manuscript very thoroughly and for giving very constructive comments. We agree regarding the comment about conclusions and we have made changes accordingly both in the abstract and in the discussion. 

1.       The authors assume causality of the observed associations and refer to influence and impact of the farming environment (starting with the title and throughout the manuscript). However, this is a small, observational and explorative study and causality cannot be implied. The wording should be changed throughout the manuscript, replacing with “association”, and the conclusiondownplayed. 

We agree with the reviewer and we have made changes accordingly throughout the paper. 

2.       Sample size: what was the rational for the inclusion of only 65 families when initially planning the study design? The authors should include a statement on the power calculation and its assumptions when designing thestudy.

The FARMFLORA birth-cohort was primarily designed to relate the maturation of the immune system to the microbiota, serum fatty acid composition and dietary pattern of farm and control children. A secondary aim was to relate allergy development to the same variables. The size of the study sample was based on the preceding ALLERGYFLORA birth-cohort, in which significant associations between continuous bacteriological and immunological parameters were observed in 60 children (Lundell, Adlerberth et al. 2007. Clin Exp Allergy 37(1): 62-71.). The sample size in the FARMFLORA birth-cohort was large enough to detect a significant difference in allergy development between Farm and non-Farm children at 3 years of age. One out of 28 Farm children (3.6%) became allergic while 10 out of 37 non-Farm children (27%) were allergic at 3 years.(Section 2.6)

3.      Line 22: “ The fatty acid pattern in cord serum is also associated with allergy development.” The statement is too strong as there is a lot of controversy. This should be changed to reflect the level of evidence. This applies also to the introduction.

We agree with the reviewer and we have made changes to this sentence.  

4.      The numbers in the abstract and methods section are not identical. It should be clear how many subjects are included in the study and how many were excluded for which reasons

The FARMFLORA birth-cohort recurited 29 farm and 37 control children. One family (farmer) withdrew early from the study due to difficulties in providing study material, resulting in 28 farm and 37 control children included in the study, of whom 2 farm and 2 control children were twins. After 1.5 year, additionally 2 families withdrew, one from each group, due to change of residence or personal issues. Biological samples were collected if possible from the infants and mothers, hence, cord blood samples, maternal serum samples and breast milk samples does not exist from all subjects. 

We have tried to make this more clear in the MS by including number of biological samples available for analysis in the method section. 

5.      The results are often overstated. It should only be reported what was exactly tested. For example in the abstract (line 29) it says “This difference could neither be explained by higher levels of n-6 PUFAs in the diet of farming mothers nor to higher n-6 PUFA proportions in their serum or breastmilk.”. As only univariate tests of fatty acid composition in cord blood, breast milk, maternal serum and diet are conducted, it cannot be implied that effects are independent. This should be reworded saying something like “differences in AA found in cord blood, but not in maternal blood, breast milk ordiet.”

We agree with the reviewer and we have made changes accordingly. 

6.      Allergies: The authors state in the abstract and discussion that allergy outcomes were not associated with fatty acid composition in cord blood. It is not mentioned and defined in the methods and results section, nor are any results presented. It is not clear which allergies, assessed how and at what time points, prevalences etc. The authors should properly analyse and present the association with allergies or delete the relevant sections in abstract anddiscussion.

We agree with the reviewer and we have added a section in the method about allergy diagnosis and we have also added the results with numbers in the result section, paragraph 3.4. 

7.      Line 75: how was tested if the cord blood sample was contaminated by maternalblood?

The cord blood was collected after delivery after the cord had been clamped and disjointed. The blood was then sampled from the placental part of the umbilical cord. We think that there is little reason to believe that the cord blood should have been contamined with maternal blood and this has therefore not been tested. 

8.      Line 113: it seems different methods have been used for the assessment of maternal diet during and after pregnancy. It is not clear, why the methods were changed as this does not allow any comparisons between time points. However,as only the diet during pregnancy was further analysed, the methods section should only focus on thisassessment.

Both dietary assessments are analyzed within the manuscript. A nutritional calculation was not performed on  the FFQ data and therefore information on specific arachidonic acid intake was taken from the nutritional assessment performed on the dietary intake measured with a 24-hour dietary recall and a prospective 24-hour food dairy one month postpartum. 

The reason to use two different assessment method is that both of them has its pros and cons. To use a FFQ is a convenient way to measure food patterns over a long time period such as the pregnancy but it is not very accurate to analyze nutritional component from an FFQ that does only give information about frequencies of intake and not information about actual amounts. 

9.      Line 124: “Wilcoxon signed rank test was used to compare maternal and infant fatty acid proportions” It is not clear where these results are presented. It would also be more clear if it would be specified which maternal measurements are meant and “cord blood fatty acid proportion” used consistently rather than “infant fatty acidproportion”

On line 124, we have missed to take away old information from a previous version of the manuscript. Regarding the last comment we have made changes in agreement throughout the manuscript. 

10.   Line 126: it is not clear when the chi2 test was applied, as all the variables seem binary orcontinuous.

Chi-2 test was used to analyze differences in characteristics between farmers and non-farmers. Example of categorical variables are parental education and parity. 

11.   Line 127: in line 122, it is explained that the variables are skewed and non- normally distributed, and therefore, non-parametric tests are applied (which is correct). Later, linear regression analysis is conducted, which is a parametric test and very prone to outliers. It should be included how the assumptions of the linear regression model were tested and if theyhold.

We agree that a linear regression is not a good test to use when having non-parametric data and we have deleted the linear regression adding butter as a confounder since butter had a very skewed distribution. Since arachidonic acid and butter is not correlated we believe that butter is not causing the higher proportions of arachidonic acid. 

12.   Line 130: Table 1 ismissing

Table 1 has been deleted and replaced with a text section. We had forgotten to change the number of table 2. This is corrected in the revised manuscript.

13.   Line 140: it is reported that “Eleven percent (11/28)” farm children were deliverd by c-section. The numbers should be double checked as 11/28 wouldbe

~40%.

The number should be 3,6  percent, i.e. 1/28. This is now changed 

14.   Table 2: the effects of PUFA sum, LCPUFA sum as well as n-6 PUFA sum and n-6 LC-PUFA sum are probably driven by AA. This should be stated at some point. 

Yes, that’s true, This is stated in the result section. 

15.   Line 178: “We have previously assessed the fatty food intake during pregnancy in the cohort [1]. Since farming mothers ate more butter, which contains some arachidonic acid, we performed linear regression model with maternal butter intake as confounder.” This statement should be included in the methodssectionbut not results. It is also not clear if the model includes only maternal butter intake during pregnancy or additionally sex and paternal education

The linear regression including butter is now deleted from the revised manuscript. 

16.   Line 194: the supplementary tables should be presented in order ofappearance in the main text. 

Yes, we have made changes accordingly. 

17.   Figure 1: change in“displayed as +/- 2 standard errors” to“mean +/- 2SD” in thelegend

The error bars are 2 standard errors. Hence, we have changed to “mean +/- 2SE”. 

18.   Line 211: “infant’s proportion… ” should be described more precisely as “proportion .. in cordblood”

We agree with the reviewer and we have made changes accordingly. 

19.   Was also assessed whether the farm mother consumed (unprocessed) farm milk duringpregnancy?

Some farm mothers consumed farm milk but this was not associated with allergy development. 

20.   Line 226: “In accordance with our findings, Donahue et al. reported strong correlations for EPA and DHA for both diet and cord plasma and maternal blood and cord plasma.” The correlations of EPA and DHA are not reported in the present manuscript therefore, the statement “in accordance with our findings” should bedeleted

We agree and have now made changes to these sentences in the manuscript. 

21.   Line 234: “nor did they have higher serum or breast milk concentrationsof these fatty acids in serum or breast-milk one month post-partum” should be corrected

Changed. 

22.   Lines 238, 239: references are not formatted correctly and should becorrected

There has been some mistake with the reference program. This is now corrected. 

23.   When discussing the potential mechanisms in the discussion section, it would be helpful to the reader to summarize all theories in one paragraph and distinguish clearly between n-6 and n-3 PUFA rather than reporting PUFA in general. This is confusing as n-6 PUFA are assumed to have a pro-inflammatory effect, as stated in line 260. The speculation on underlying mechanisms in the conclusion should be deleted (lines274-276)

We have revised the discussion and separated the role of n-6 and n-3 PUFA. Only arachidonic acid forms eicosanoids e.g. prostaglandins. However both n-3 PUFAs and n-6 PUFAs are strongly immunosuppressive in that they reduce T-cell activation and signaling as well as production of interferon gamma by T-cells. We mention the possible role of prostaglandins (which are derived from the n-6 PUFA arachidonic acid) and have added another possible explanation in response to reviewer 2.

We have also added one sentence in the result section demonstrating that total n-6 and n-3 PUFAs in cord blood correlated significantly as did arachidonic acid and DHA in cord blood.

We would preferably keep the speculation in the conclusion as we want to give some possible explanations for the observed associations.

Round 2

Reviewer 3 Report

I thank the authors for their revision. However, not all comments have been adequately addressed. 

Additionally, it would be very much appreciated if the authors use track mode when preparing their revision and include line numbers and citations in their response letter. As it is now, it is nearly impossible and extremely time consuming to compare the two pdf files and detect the changes made. 

Regarding comment 2: The authors included the requested information on the study design in the response letter, but I could not find how this was incorporated in the main text. Furthermore, there is no description of the power calculation.

Regarding comment 6: It is stated that allergic diseases were based on a combination of several symptoms and diagnosis. For the definition, the authors refer to reference 1. However, a brief explanation that e.g. asthma is wheezing symptoms would be helpful.  

What is the reason to exclude the children with allergies at 18 months only?  Eczema is included only when it develops after the age of 2 years according to the definition in the reference publication, but it would be plausible to see effects in early eczema when focusing on the cord blood composition.   

There is only 1 child with allergies from a farm. I agree that this is substantially lower than the 10 children in the non-farm group. However, the number of allergic children is too low for analysis and to draw any sound conclusions, especially in relation to the farm background.

Regarding comment 10: The chi2 test is an approximate test (e.g. https://en.wikipedia.org/wiki/Exact_test). The numbers in the cells might be too low. It is recommended to use exact tests instead. This would be the exact Fisher test (for 2x2 tables) or the exact version (if no ties) of the Wilcoxon rank sum test (Mann-Whitney U test) for continuous variables. I could not find parity or parental education in the manuscript. For paternal education, there is no information on its distribution or definition. 

Regarding comment 11: There are two different problems the authors are referring to. The first is related to the assumptions of linear regression, which is that the errors are required to be normally distributed. This indirectly implies the normal distribution of the depended variable (which would be arachidonic acid concentrations here). More information on the assumptions and how to test them in SPSS can be found e.g. https://stats.idre.ucla.edu/spss/seminars/introduction-to-regression-with-spss/introreg-lesson2/

It is not required that the continuous independent variables (e.g. butter consumption) follow normal distribution. However, in linear regression, it is assumed that the effect of the independent variable on the dependent variable is linear (hence the name). If the independent variable is very skewed, it does not necessarily mean that the effect is not linear. The linearity should also be checked, e.g. visually with a scatter plot and a non-linear smoother compared to a univariate regression line. 
